# Leaf Lipid Alterations in Response to Heat Stress of *Arabidopsis thaliana*

**DOI:** 10.3390/plants9070845

**Published:** 2020-07-04

**Authors:** Sunitha Shiva, Thilani Samarakoon, Kaleb A. Lowe, Charles Roach, Hieu Sy Vu, Madeline Colter, Hollie Porras, Caroline Hwang, Mary R. Roth, Pamela Tamura, Maoyin Li, Kathrin Schrick, Jyoti Shah, Xuemin Wang, Haiyan Wang, Ruth Welti

**Affiliations:** 1Division of Biology, Kansas State University, Ackert Hall, Manhattan, KS 66506-4901, USA; thilani.samarakoon@gmail.com (T.S.); kaleb.a.lowe@Vanderbilt.edu (K.A.L.); drewroachcar@gmail.com (C.R.); vusyhieu@gmail.com (H.S.V.); mcolter@kumc.edu (M.C.); hollieporras11@gmail.com (H.P.); carolineh0205@ku.edu (C.H.); mrroth@ksu.edu (M.R.R.); ptamura@ksu.edu (P.T.); kschrick@ksu.edu (K.S.); 2Department of Biological Sciences, University of North Texas, Denton, TX 76203-5017, USA; maoyinli@hotmail.com (M.L.); SWang@danforthcenter.org (X.W.); 3Department of Biology, University of Missouri-St. Louis, St. Louis, MO 63121-4499, USA; 4Donald Danforth Plant Science Center, St. Louis, MO 63132, USA; shah@unt.edu; 5Department of Statistics, Kansas State University, Manhattan, KS 66506-0802, USA; hwang@ksu.edu

**Keywords:** *Arabidopsis thaliana*, heat stress, lipidomics, oxidized lipids, acylated lipids, phosphatidic acid, polygalactosylated lipids, triacylglycerols, sterol glucosides, acyl sterol glucosides

## Abstract

In response to elevated temperatures, plants alter the activities of enzymes that affect lipid composition. While it has long been known that plant leaf membrane lipids become less unsaturated in response to heat, other changes, including polygalactosylation of galactolipids, head group acylation of galactolipids, increases in phosphatidic acid and triacylglycerols, and formation of sterol glucosides and acyl sterol glucosides, have been observed more recently. In this work, by measuring lipid levels with mass spectrometry, we confirm the previously observed changes in *Arabidopsis thaliana* leaf lipids under three heat stress regimens. Additionally, in response to heat, increased oxidation of the fatty acyl chains of leaf galactolipids, sulfoquinovosyldiacylglycerols, and phosphatidylglycerols, and incorporation of oxidized acyl chains into acylated monogalactosyldiacylglycerols are shown. We also observed increased levels of digalactosylmonoacylglycerols and monogalactosylmonoacylglycerols. The hypothesis that a defect in sterol glycosylation would adversely affect regrowth of plants after a severe heat stress regimen was tested, but differences between wild-type and sterol glycosylation-defective plants were not detected.

## 1. Introduction

Global climate change is increasing heat-induced plant stress, which can affect crop growth, quality, and productivity. Among the changes plants make in response to exposure to high temperature are those in lipid metabolism. 

The maintenance of membrane fluidity and permeability can influence plant ability to survive and flourish. A decrease in unsaturation of leaf fatty acids in membrane lipids is one lipid metabolic response that has long been known to occur in response to high temperature [1,2,3,4,5]. Reduced desaturation of newly synthesized fatty acids and increased lipid turnover are likely to contribute to the reduction in unsaturation, which may help to maintain appropriate membrane fluidity at high temperature. 

Recent work on plant response to heat points to further lipid changes and their enzymatic basis. Chief among these are increases in leaf triacylglycerol (TAG) species, particularly those containing 18:3, under heat stress [5,6,7,8]. Mueller et al. [7] demonstrated that accumulation of leaf TAGs in Arabidopsis is dependent, at least in part, on the activity of phospholipid: diacylglycerol acyltransferase 1 (PDAT1), which can transfer an acyl chain from phosphatidylcholine (PC) to diacylglycerol (DAG) to form TAG. An increase in 16:3-containing lipid species, including PCs, DAGs, and TAGs, also occurs during heat stress. In addition, levels of acylated monogalactosyldiacylglycerols (acMGDGs) and polygalactosylated galactolipids (tri- and tetra- galactosyldiacylglycerols; TrGDGs and TeGDGs) rise in response to heat [7].

Increases in levels of sterol glucosides (SGs) and acyl sterol glucosides (ASGs) occur when plants are under high-temperature stress [8,9]. Plants synthesize a mixture of sterols, including sitosterol, stigmasterol, and campesterol. SGs are synthesized by sterol glucosyltransferases, which catalyze formation of a glycosidic bond between a carbohydrate moiety (typically glucose) and the free hydroxyl group on a sterol molecule. SGs may be acylated on the 6-position of glucose to form ASG. While the biological significance of increased SG and ASG under heat stress needs further investigation, recent work hints at a possible role in improving response to heat [8,9].

Oxidation of membrane lipids occurs in response to both biotic and abiotic stresses, including pathogen infection, wounding, and freezing [10,11,12]. While previous work in wheat identified several oxidized PC and phosphatidylethanolamine (PE) species induced under heat stress [8], little is known about the production of oxidized chloroplast membrane lipids.

In the current work, direct-infusion electrospray ionization (ESI) triple quadrupole mass spectrometry, employing a quality control strategy to enhance precision [13], was utilized to comparatively profile lipids of *Arabidopsis thaliana* leaves through a time course that incorporates heat stress. We confirm observations of multiple changes reported previously. In addition, we report that Arabidopsis leaves exhibit increased levels of plastidic lipids with oxidized fatty acyl chains, monoacyl galactolipids, and sterol derivatives in response to heat stress. We used UDP-glucose:sterol glucosyltransferase mutants to further investigate the role of SGs and ASGs under controlled conditions, but no differences in plant recovery from severe heat stress were detected.

## 2. Results and Discussion

### 2.1. The Main Experiment Involved a Moderate Heat Stress Treatment

In our main experiment, *Arabidopsis thaliana* (Columbia-0 accession) plants, grown at 21 °C for 28 days in a 14/10 light/dark cycle, were subjected to one of three treatments: “Path 1”) control treatment of 40 additional hours at a growth temperature of 21 °C; “Path 2”) heat treatment of 12 h at 38 °C, 4 h at 45 °C, and 24 h at 21 °C; or “Path 3”) 12 h at 21 °C, 4 h at 45 °C, and 24 h at 21 °C (Figure 1). Plants were maintained in their 14/10 light/dark cycle during the treatments, which started 1 h before the onset of the dark phase. We planned these treatments so that the 38 °C treatment in Path 2 would serve as an acclimation for the 45 °C treatment in that path, whereas Path 3 would represent heat treatment without acclimation. However, we have no evidence that the 38 °C treatment did result in acclimation, so the main experimental treatments are referred to only as Path 2 and Path 3 in this report, and the control treatment is Path 1. Neither of the two heating treatments caused leaf death, and the plants exhibited no detectable wilting 24 h after the heat treatment (Appendix A). A recent paper also found that wild-type Arabidopsis survived treatments of 45 °C and 38 °C, followed by 45 °C, albeit for shorter time periods [14]. Measurements of ion leakage on six plants at each time point (others not measured) did not indicate an increase in ion leakage due to the heat treatment, as would have been expected if the heat treatment caused cell lysis (Appendix A). The main phenotype observed with heating was raising of the leaves, which occurred to the largest extent during the 12 h 38 °C treatment in Path 2, which included the dark period of the light cycle (Appendix A). Leaf raising that occurred during the 38 °C treatment of Path 2 dissipated only slightly during the 4 h 45 °C treatment and somewhat more over the 24 h recovery period (Appendix A). Relatively slight leaf raising was visible at the end of the 45 °C heat treatment of Path 3 (Appendix A) but dissipated after the 24 h recovery period (Appendix A). Leaf raising in relation to heat is known as thermonasty, an auxin-mediated response that helps to cool the leaves [15,16]. Arabidopsis also raises its leaves in its daily cycle, but this diurnal leaf raising begins at dawn [17], and the raising of the leaves in the current work was also observed during the dark period at 38 °C. Thus, the leaf raising in Path 2 is likely to be associated with exposure to higher temperatures.

### 2.2. Leaf Lipid Levels Were Determined as a Function of Heat Treatment

The rosettes of 18 plants were harvested at each time point shown in Figure 1. Each rosette was extracted and analyzed separately, and 277 lipid molecular species were measured by direct-infusion electrospray ionization triple-quadrupole mass spectrometry with multiple reaction monitoring (MRM) [13]. The lipid analytes and their acquisition parameters are listed in Appendix A. The analytical data were normalized to quality control samples, and limit of detection (LOD) and coefficient of variation (CoV) were assessed for each lipid analyte, as described previously [13]. Most (247 or 89%) of the lipid analytes met a CoV criterion of <0.2 and an LOD criterion of >0.25 units of signal intensity. Information on the quality of the data for individual analytes is shown in Appendix A, and all data for lipid levels are provided in Appendix A. Mass spectral intensities were normalized to levels of internal standards, which are specified for each analyte in Appendix A. In some cases, due to lack of availability of appropriate internal standards, the structures of the internal standards were not well-matched to those of the compounds measured. Thus, while comparison of any lipid analyte across samples is valid, comparison of amounts of analytes with each other is not accurate, because response factors for each compound to its internal standard were not determined or applied. 

In Figure 2a, levels of each major lipid class at the end of the heating treatments in Paths 2 and 3 are shown, relative to the level in Path 1 (the control). The head group class data and the statistical results are provided in Appendix A. ANOVA indicated that, at the 16-h (end of heating) time point, after 12 h at 38 °C and 4 h at 45 °C in Path 2, levels of acMGDG, digalactosylmonoacylglycerol (DGMG), lysophosphatidylcholine (LPC), monogalactosylmonoacylglycerol (MGMG), phosphatidic acid (PA), SG, ASG, TAG, and combined TrGDG and TeGDG were higher than at the same time point in Path 1 (21 °C). The levels of phosphatidylethanolamine (PE) and phosphatidylglycerol (PG) were modestly lower after the heat treatment in Path 2 than in Path 1 plants, whereas sterol ester levels, which are low in leaves in all circumstances, were only 26% of the level in Path 1. Changes in Path 3 tended to be similar to those in Path 2, but not as pronounced (Figure 2a). The levels of the lipid classes that were significantly increased at the 16 h time point of Path 2 are shown across the time course in Figure 2b. The heat map indicates that the increases in these heat-induced lipid classes were maximal at end of the heating treatment and that the increases were transient.

The levels of individual lipid molecular species in leaves of plants undergoing the three treatments were also compared at each time point. One hundred forty-four lipid species were significantly different, after correction for false discovery rate (FDR), in one or both heat treatments (Path 2 or 3) compared to the control treatment (Path 1) at one or more time points. The greatest number of analytes (92) was significantly different from control at the 16-h time point, which marked the end of the heating treatment. Thirteen analytes differed from the control at the 1 h time point, 15 analytes at the 12 h time point, 30 analytes at the 13 h time point, 8 analytes at the 17 h time point, and 35 analytes at the 40 h time point. The fold differences of significantly changed lipid analytes at 16 h are shown in Figure 3, Figure 4 and Figure 5, and the lists of analytes altered at all time points are reported in Appendix A. 

### 2.3. Moderate Heat Treatment Induced Leaf Lipid Acylation and Oxidation of Chloroplast-Localized and Extraplastidically Localized Lipid Species

Head group acylation and fatty acyl oxidation of leaf plastidic lipids (monogalactosyldiacylglycerol (MGDG), digalactosyldiacylglycerol (DGDG), and PG) occur commonly when Arabidopsis is exposed to stresses that include freezing, wounding, and pathogen infection [10,11,12]. Leaf PC and PE, which are primarily localized outside of the plastid, also display oxidized fatty acids during wounding, and pathogen infection [11,13]. Fatty acylation of MGDG on the 6-position of the galactose ring was first demonstrated by Heinz [18,19]. In our previous work [11,12,20] and in the present study, oxidized fatty acyl chains are designated by indicating the number of carbons, number of double bond equivalents, and number of oxygens in addition to the carbonyl oxygen. For example, oxophytodienoic acid (OPDA), a fatty acid derived from linolenic acid and containing a cyclopentenone ring, contains 18 carbons, four double bond equivalents (two C-C double bonds, the ring, and the ketone), and one oxygen (the ketone) in addition to the carbonyl oxygen; thus the detected chain is abbreviated 18:4-O [20]. The identities of fatty acids consistent with the acyl chain chemical formulas characterized by mass spectrometry have been summarized previously (Table S7 of ref. [13]). 

In Arabidopsis, plastidic lipids are head group-acylated with oxidized, as well as non-oxidized, fatty acids. Evidence indicating the presence of over 60 acylated (mostly acMGDG) and more than 50 diacyl galactolipids and phospholipids, containing oxidized chains, has been summarized (Table S5 of ref. [13]). The common oxidized fatty acids, OPDA (18:4-O) and its 16-carbon analog, dinor-OPDA (dnOPDA, 16:4-O), are linked to the 1- and 2- positions of glycerol in both acylated and non-acylated galactolipids, and to the galactose ring in acMGDG [10,20,21,22]. Collectively, the molecular species of galactolipids composed of only OPDA or dnOPDA fatty acids are called Arabidopsides. Arabidopside A (ArA) is MGDG(OPDA/dnOPDA) or MGDG(18:4-O/16:4-O), ArB is MGDG(18:4-O/18:4-O), and ArD is DGDG(18:4-O/18:4-O) [21,22]. ArE is ArA, head group-acylated with OPDA (i.e., acMGDG(18:4-O/34:8-2O), where 34:8-2O indicates the combination of fatty acids in the DAG portion (i.e., 18:4-O/16:4-O)) [10,23]. ArG is the all 18-carbon-acylated version, i.e., acMGDG(18:4-O/36:8-2O) [23]. 

Acylation of the MGDG head group (galactose) is catalyzed by the enzyme encoded by *At2g42690*; the enzyme has been named acylated galactolipid associated phospholipase 1 (AGAP1) [24]. AGAP1 transfers a glycerol-linked fatty acyl chain from DGDG or MGDG to the 6-position of the galactose ring of MGDG, producing a monoacyl galactolipid, as well as an acMGDG. The monoacyl product can be MGMG or DGMG, but previous work demonstrates a preference for DGDG as an acyl donor [19,25]. AGAP1 is responsible for acylation of MGDG during freezing stress and in the hypersensitive response of plants to pathogenic bacteria [10,11,23]. Mueller et al. [7] identified normal-chain acMGDGs that increase in prominence during heating. The two major species reported were acMGDG(54:9) and acMGDG(52:9), which represent species with three 18:3 chains and an 18:3/18:3/16:3 combination. Figure 3 confirms that heat stress induces increased levels of four normal-chain acMGDG species. The observed molecular species all contain a 34:6 (18:3-16:3) DAG backbone and vary in the head group-acylating fatty acids. (Note that, in the present study, mass spectrometry detected the mass of a neutral loss of an acyl-containing, head group-specific fragment, as well as the intact ion mass/charge (*m*/*z*), providing clarity on which acyl species in acMGDG is linked to the galactose.) Among the normal-chain species, at the 16 h time point in Path 2 (12 h treatment at 38 °C followed by 4 h treatment at 45 °C), compared to Path 1, acMGDG(16:3/34:6) was increased the most (approximately 5-fold) by heat.

In addition to normal-chain species, as shown in Figure 3, heat induced increases in numerous oxidized acMGDG species, including those with oxidation on the head group and diacylglycerol fatty acyl chains. The oxidized chains in acMGDG include 16:3-O, 18:3-O, and 18:2-O, which are likely hydroxy fatty acids, 18:4-O (OPDA), and likely 16:4-O (dnOPDA, as a component of the DAG backbone annotated as 34:8-2O, which corresponds to OPDA/dnOPDA). Additionally, 18:3-3O and 18:4-2O in acMGDG were detected and are consistent with phytoprostane structures, while 18:4-3O and 18:5-2O are unknown structures [13]. The greatest fold increase (about 4-fold) for an oxidized acMGDG occurred in acMGDG(18:3-O/34:8-2O) (or acMGDG(18:3-O/36:6)). This species has an oxidized (hydroxy) fatty acid on the galactose ring. The identity of the diacyl species is ambiguous, and could be OPDA/dnOPDA(34:8-2O or 18:4-O/16:4-O), 18:3/18:3 (36:6), or a combination of these, since these alternative diacylglycerol components have the same nominal mass and are not differentiated by direct-infusion ESI triple quadrupole mass spectrometry. Time courses in Figure 6a,b show that the two most highly altered acMGDG molecular species (one containing two normal chains and one with at least one oxidized chain) tended to increase when plants were exposed to 12 h at 38 °C or 4 h at 45 °C, but increased much more when the 38 °C and 45 °C treatments were combined. 

Figure 4 shows plastidically-formed di- and monoacyl lipids associated with heat stress at the 16-h time point of Paths 2 and 3. Approximately 3- to 4-fold increases in DGMG(18:4-O) (time course shown in Figure 6c) and MGMG(16:0) were observed, along with a smaller increase in MGMG(18:4-O). A rise in DGMG and MGMG is consistent with the formation of acMGDG by AGAP1, since these species are also generated when AGAP1 transfers a fatty acid from DGDG or MGDG to the MGDG head group. In contrast, MGMG increases were not detected by Mueller et al [7], who hypothesized that these “lyso-galactolipids” might be rapidly turned over in 2-week-old plants subjected to heat treatment. It is conceivable that this difference relates to the age of the plants, as the plants used here were 28 days old at the start of the treatment. It is also possible that MGMG and DGMG are generated by the direct action of acylhydrolases on MGDG and DGDG, in addition to the AGAP1 activity.

Besides acMGDGs, there were increases in other chloroplast-localized lipid species with oxidized fatty acyl chains during heat stress. Oxidized DGDGs, MGDGs, PGs, and sulfoquinovosyldiacylglcyerols (SQDGs) were observed (Figure 4). Oxidized DGDGs include 5 molecular species containing 18:4-O (OPDA), with the highest fold increase (4- to 5-fold) seen in DGDG(18:4-O/18:4-O) or Arabidopside D. Similarly, OPDA is prominent in oxidized MGDG molecular species, with 11 heat-induced species containing OPDA (18:4-O) and three containing dnOPDA (16:4-O). Three oxidized, heat-induced PG species contain OPDA. While the fatty acyl chains in SQDG were not individually detected, the combinations are consistent with SQDG (16:0/18:4-O) and SQDG (18:4-O/18:4-O). Arabidopside D and PG(18:4-O/16:1) in plants treated for 12 h at 21 °C plus 4 h at 45 °C or 12 h at 38 °C plus 4 h at 45 °C had similar time courses, peaking at the end of the 45 °C treatment, with no significant increase during 38 °C heating (Figure 6d,e). 

There is evidence indicating that the enzymatic conversion of 18:3 to 18:4-O (OPDA), which includes four reactions occurring in the chloroplast, catalyzed by lipoxygenase, allene oxide synthase, allene oxide cyclase, and oxophytodienoic acid reductase, can take place on intact glycerolipids, and not solely on free fatty acids [26]. Indeed, the oxidized chloroplast-localized molecular species formed under heat stress are consistent with oxidation of acyl chains directly on the diacylglycerol-containing lipids of plastids. The most common DGDG, DGDG(18:3/18:3), is oxidized to Arabidopside D and the most common MGDG, MGDG(18:3/16:3), to Arabidopside A. The most common PGs, PG(18:3/16:1) and PG(18:3/16:0) are oxidized to the OPDA-containing versions, PG(18:4-O/16:1) and PG(18:4-O/16:0). The observed oxidized SQDG species are consistent with conversion of 18:3 to OPDA on the most common SQDG species (SQDG(16:0/18:3) and SQDG(18:3/18:3)). Supporting the notion of OPDA and dnOPDA formation occurring enzymatically, Appendix A shows that a reduction in the activity of oxophytodienoic acid reductase tends to lower the formation of Arabidopsides by 50 to 70%. The mutant employed in these studies is not the classic *opr3* mutant [27], but a line, obtained from Jianmin Zhou, containing a defective version of this gene [28]. 

Heat significantly increased levels of two PCs and four PEs with oxidized fatty acyl chains, although the fold increases were less than 2-fold (Figure 5). Similarly, levels of the two monitored *N*-acyl PE (NAPE) species were slightly increased by heat treatment (Figure 5). While other stresses, including freezing and wounding, induced production of oxidized PC and PE species, the fold increases of these extra-plastidically assembled oxidized lipids occurring in response to heat treatment were generally lower in comparison to chloroplast-localized oxidized lipid species [11].

### 2.4. Galactolipid Polygalactosylation and Increased Levels of PCs and PEs Containing a 16:3 Acyl Chain Were Observed after Moderate Heat Treatment

SENSITIVE TO FREEZING 2 (SFR2, the product of *At3g06510*) is an enzyme that catalyzes the transfer of a galactose moiety from one MGDG to another MGDG or to a growing DAG-linked polygalactose chain [29,30]. Besides a polygalactosylated DAG, the other product of this enzyme is DAG. SFR2 was first described based on its role in response to freezing, in which its activity confers increased freezing tolerance [31,32]. However, SFR2 activity is also associated with wounding and heat treatment [7,33]. An increase in TrGDG(34:6) was detected at the 16 h time point in both heat treatments (Figure 4). Although little increase in TrGDG(34:6) occurred during the 38 °C treatment, the fold increase upon the 45 °C treatment tended to be higher when the plants were first subjected to 38 °C treatment (Figure 4 and Figure 6f). 

Increases in PC and PE molecular species containing 16:3 were also observed (Figure 5). Appearance of 16:3, which is synthesized in the plastid, in extraplastidically localized phospholipids at elevated temperature was observed previously [7,34]. One possible source of the increased 16:3 observed in PC and PE species, including PC(32:3), PC(34:6), and PE(32:3), upon heat treatment, is the DAG released from MGDG when polygalactosylated DAGs (e.g., TrGDG and TeGDG) are formed by SFR2. However, the time course for formation of these species (Figure 7a) differs somewhat from that for TrGDG(34:6) (Figure 6f). Moreover, in MGDG and DGDG, 16:3 also occurs only in combination with an 18-carbon fatty acid. In PC and PE, 16:3 occurs in combination with 16:0. This 16:0/16:3 (32:3) combination is not observed in MGDG or DGDG and suggests that at least some 16:3 is released by hydrolysis from MGDG, DGDG, or plastid-generated DAG before incorporation into PC and PE. 

### 2.5. Heat Treatment Induced Increased Levels of Phosphatidic Acid

Levels of several major leaf PA molecular species increased roughly 2-fold upon heat treatment (Figure 5). The time course of PA(34:2) levels is shown in Figure 7b. The increase in PA was not affected by exposure of plants to 38 °C treatment. Previous work by Mishkind et al. [35] in suspension-cultured tobacco cells and in Arabidopsis and rice seedlings indicated that heat stress leads to formation of PA by phospholipase D (PLD), rather than by phospholipase C (PLC) followed by DAG kinase. Recently, heat stress has been shown to activate PLDδ, which has a negative effect on plant thermotolerance via destabilizing cortical microtubules [36]. On the other hand, Arabidopsis non-specific phospholipase C1 (NPC1) also has been shown to be involved in basal thermotolerance, but NPC1 is expressed mostly in roots [37]. Two phosphoinositide-specific PLCs also have been demonstrated to play roles in heat tolerance [38], but it is likely that the amount of PA originating from DAG formed by these enzymes is small. Certainly, the PA molecular species, PA(34:2), PA(34:3), PA(36:2), and PA(36:3), which were significantly increased by 16-h treatments, are consistent with an origin of the PA in extra-plastidically localized phospholipids. The production of these molecular species is also consistent with PA resulting from the activity of PLD, and, in particular, PLDδ, which is typically plasma membrane-localized, and which increases levels of PA(34:2) and PA(34:3) when overexpressed [39].

### 2.6. Moderate Heat Treatment Increased Leaf Triacylglycerol Levels 

Figure 2 and Figure 5 show that levels of triacylglycerols were increased strongly when plants were subjected to heat stress in either Path 2 or 3. Most of the significantly altered TAG species increased 8- to 9-fold, while the 16:3-containing TAG species, TAG(18:3/34:6), i.e., TAG(18:3/18:3/16:3) or TAG(52:9), was over 30-fold higher in Path 2, compared to Path 1. Mueller et al. [6,7] found that TAGs accumulated in 14-day-old Arabidopsis plants in both shoots and roots when the plants were exposed to heat stress. The species identified as changing significantly here include four of the five molecular species that Mueller et al. [6] observed accumulating at the highest levels. Similar to our findings, TAG(52:9) showed the largest fold increase, both at 38 °C and 45 °C [6,7]. Mueller et al. [6] demonstrated that TAG accumulation occurs primarily outside of the chloroplasts. Higashi et al. [5] observed an increase in TAG(18:3/36:6), i.e., TAG(18:3/18:3/18:3) or TAG(54:9), in 14-day-old plants subjected to 38 °C heat stress. TAG(18:3/36:6) was also one that we found to be significantly altered (Figure 5). Time courses showing the accumulation of TAG(18:3/34:6) and TAG(18:2/36:5), i.e., TAG(18:2/18:2/18:3), indicate that both TAGs increase significantly at 38 °C, and increase further at 45 °C in Path 2 (Figure 7c,d). 

Mueller et al. [7] demonstrated that seedlings deficient in phospholipid:diacylglycerol acyltransferase 1 (PDAT1) had much lower accumulation of TAGs under heat stress, compared to wild-type plants, pointing to the importance of PDAT1 in TAG accumulation under heat stress. PDAT1 catalyzes the transfer of a fatty acid from PC to DAG to form TAG and LPC. Mueller and coworkers [7] suggested that molecular species of PC that increase during heat stress serve as a fatty acyl donors for PDAT1’s acylation of DAG. Higashi et al. [40] identified a heat inducible lipase that releases the 18:3 from the 1-position of MGDG and contributes to the formation of TAG during heat stress. This, coupled with evidence for involvement of PDAT1, suggests that fatty acids from MGDG must move to TAG through PC and/or DAG. However, because 16:3, as well as 18:3, moves to TAG, and since 16:3 is found only in the 2-position of MGDG, an additional enzyme (an *sn*-2 lipase) is likely to be involved in fatty acid release from MGDG. An acyltransferase, perhaps associated with the chloroplast outer membrane, may transfer the released fatty acid to PC. Recently, lysophosphatidylcholine acyltransferasea (LPCAT1 and LPCAT2) were found to be associated with the outer chloroplast membrane and to account for most of the chloroplast-associated LPCAT activity [41]. The source of the DAG moiety used in the TAG synthesis is not clear. Mueller et al. [7] determined that it is not derived from the action of phosphatidylcholine:diacylglycerol phosphocholine phosphotransferase (PDCT, encoded by *ROD1*, *At3g15820*). Instead it could be derived from PC by CDP-choline:diacylglycerol cholinephosphotransferase (CPT) activity acting in the reverse direction (PC → DAG + CDP-choline), from PC by lipase activity, from de novo synthesis by the Kennedy pathway, from the galactolipid polygalactosylation pathway (which produces DAG as a byproduct), from another galactolipid-hydrolyzing pathway, or from some combination of activities [7,42].

### 2.7. Unsaturation Indices of Major Diacyl Lipid Species Decreased and Those of Triacylglycerols Increased under Moderate Heat Treatments

Among the molecular species significantly increased at the 16-h time point of the heat-stress regimens were MGDG(36:3) and PC(32:0) (Figure 4 and Figure 5). Each of these species are less unsaturated than the major molecular species of their classes. A time course showing levels of MGDG(36:3) (Figure 7e) indicates that the rise in MGDG(36:3) is not fully reversed after 24 h at 21 °C. The observation of increased levels of these species led us to calculate the unsaturation indices (Figure 8 and Appendix A). Heat reduced the double bonds in both PC and PE molecular species in both Paths 2 and 3, compared to Path 1. DGDG, MGDG, and PS had significantly lower unsaturation indices at the 16 h time point of Path 3, compared to Path 1. In contrast, the TAG pool was more unsaturated than control samples at the 16 h time point of both heat conditions. These data, combined with previous observations, indicate that not only does the TAG pool increase in size, but it increases in unsaturation in response to heating. The combined data support the notion that the TAG pool is a sink for unsaturated fatty acids removed from membrane lipids during heat stress.

Chain length indices, calculated similarly to unsaturation indices, were also analyzed for the major lipid classes (Appendix A). All chain length differences were small, but analyzed molecular species of MGDG, PS, and TAG had longer fatty acid chains under one or both heat stress treatments, while PA, PC, PE, and PG had significantly shorter chains. The significant decreases in the latter classes may seem, on the surface, unexpected, since higher temperatures typically call for organisms to put more rigid fatty acids in their membranes, and longer chains with the same number of double bonds are more rigid than shorter chains. However, the classes with decreased chain length are very likely to incorporate 16:0 into their membranes to replace 18-C polyunsaturated fatty acids, leading us to conclude that the effects on acyl-chain length are likely secondary to changes in unsaturation. 

### 2.8. Sterol Glucoside and Acyl Sterol Glucoside Levels Increased with Heat Treatment

SGs and ASGs are major components of plant plasma membranes [43,44,45]. SGs are formed when glucose is transferred to a sterol from UDP-glucose by a UDP-glucose:sterol glucosyltransferase (UGT). UGT80B1 (product of *At1g43620*) and UGT80A2 (product of *At3g07020*) catalyze the formation of over 85% of the sterol glucosides in Arabidopsis leaves [46]. Addition of a fatty acid to the 6-position on the glucose of a sterol glucoside by an unknown acyltransferase forms an ASG. Heating increased the levels of SGs and ASGs containing campesterol, sitosterol, and stigmasterol, the three most abundant Arabidopsis sterols (Figure 2 and Figure 5). A time course showing levels of Sitosterol-Glc(18:3) is shown in Figure 7f.

### 2.9. Reduction in Sterol Glucosides and Acyl Sterol Glucosides Did Not Impact Plant Survival and Growth after Severe Heat Treatment

In sorghum, levels of SGs and ASGs increase during heat stress [8]. Comparison of lipid composition of a heat-tolerant sorghum cultivar and a heat-susceptible sorghum cultivar identified differences in levels of SGs and ASGs under heat stress, with the heat-tolerant cultivar displaying higher levels of sterol derivatives, consistent with the notion that SGs and ASGs might play a role in improving plant response to heat [8]. Singh et al. [47] showed that silencing of the genes for three glycosyltransferases that form sterol glycosides in the medicinal plant, *Withania somnifera*, reduced the photosynthetic rate and increased the transpiration rate of the plants under heat stress. Furthermore, *ugt80B1* mutants of Arabidopsis, in which SG content is reduced by 65–80%, exhibited decreased survival of seedlings grown on nutrient agar medium for 5 days at 42 °C [9]. 

Our group tried repeatedly to test the hypothesis that SGs and ASGs play a role in growth or survival of Arabidopsis under heat stress. In our experimental design, 30-day-old double mutants of *UGT80A2* and *UGT80B1* [47] and wild-type plants were subjected to severe heat stress, 45 °C for 12 h, or a control treatment (21 °C). This 45 °C treatment was three times longer than for Path 3 in the experiment described in Figure 1. The severe heat stress did not kill the plants but caused the outer rosette leaves to wilt and eventually to die. Appendix A shows 30-day-old plants before and after the heat treatment, and after 12 days of regrowth. Also shown are 42-day-old control plants (Appendix A, maintained at the 21 °C growth temperature). 

At the end of the 12 h heat or control treatment, a wilted leaf (leaf 4), which would have later died in the treated plants, was sampled and the lipid composition determined by direct-infusion electrospray ionization triple–quadrupole mass spectrometry with multiple reaction monitoring (MRM), as previously performed for the main experiment (moderate heat stress). The compositional data for the plants under severe heat stress are displayed in Figure 9 and Appendix A. Under severe heat stress, in leaf 4, the combined SG level in the *ugt80A2,B1* double mutants was 78% lower than that of wild-type plants (Figure 9). This percent reduction under heat stress is similar to that observed for the double mutant under normal growth conditions (Figure 9 and [46,48]). Both wild-type plants and the *ugt80A2,B1* double mutants exhibited increases in the same lipids previously observed to increase under moderate heat stress (acMGDGs; DGMGs; MGMGs; oxidized galactolipids, SQDGs, and phospholipids; LPCs; PAs; TAGs; polygalactosylated diacylglycerols; SGs and ASGs) (Figure 9 and Appendix A). However, the fold increases of some classes in wild-type plants, particularly TAGs and polygalactosylated galactolipids, were greater in the severe 12 h 45 °C heat stress, compared to either the Path 2 or Path 3 treatments of the moderate heat stress (Figure 9 compared to Figure 2; Appendix A compared to Appendix A). Additionally, with the longer heat treatment, DAGs and lysoPEs (LPEs) were also significantly increased, and levels of many normal-chain membrane lipids, e.g., DGDGs, MGDGS, PCs, PEs, PGs, PIs, PSs, and SQDGs, were strongly decreased (Figure 9 and Appendix A). Not only were SGs and ASGs lower in *ugt80A2,B1* double mutants than in wild-type plants, but some acMGDGs and PAs were higher in the heat-treated double mutants compared to those in wild-type plants (Figure 9, Appendix A). Additionally, sterol esters, which are present in low amounts, were higher in leaves of the *ugt80A2,B1* double mutants compared to wild-type plants, both before and after the heat stress (Figure 9 and Appendix A), suggesting that a deficit in the sterol glycosylation pathway may increase fatty acylation of sterols.

Despite the observed differences in lipid composition between leaves of wild-type and *ugt80A2,B1* double mutants, no alteration in growth pattern was apparent in our experiment. While we observed some variability in results of small-scale, preliminary experiments, the large experiment did not reveal significant growth differences between wild-type plants and the *ugt80A2,B1* double mutants subjected to the 12 h, 45 °C heat stress applied to 30-day-old, soil-grown plants (Table 1 and Table 2). Table 1 shows the number of living leaves after plants recovered for 11.5 days, following the heat treatments, whereas Table 2 shows rosette weight at the same time points. Both sets of data indicate that there is no detectable difference in vegetative growth and regrowth of the *ugt80A2,B1* double mutants compared to wild-type plants after the 12 h heat stress. Besides the obvious interpretation that SGs and ASGs are not required for reducing damage or hastening recovery after heat stress, it is possible that SGs are less important in older plants compared to seedlings, which Mishra et al. [9] examined when they observed survival differences between *ugt80B1* mutants and wild-type plants in response to extended heat stress at 42 °C. It is also possible that very specific heating conditions are required to observe differences due to SGs and ASGs or that other parameters may be more sensitive measures of the effects of SGs and ASGs than vegetative growth. It would be an interesting future avenue to test the effect of SG and ASG levels on plant reproduction, as there is abundant evidence that plant reproductive structures, and particularly pollen, are among the plant components most sensitive to heat [49]. 

## 3. Materials and Methods 

### 3.1. Plant Materials

Wild-type *Arabidopsis thaliana* was accession Col-0. The *opr3* line was kindly provided by Jianmin Zhou. It was identified from an EMS mutagenized population of a transgenic RAP-luciferase line (in Col-0 background) that had a G2471A base substitution in *OPR3* resulting in replacement of Trp138 by a stop codon. This line is not male-sterile and was used previously by Cheng et al. [28]. The *ugt80A2,B1* double mutant in the Col-0 background was described by Stucky et al. [50]. 

### 3.2. Plant Growth

Pro-Mix “PGX” soil (Hummert International, Earth City, MO, USA) was mixed with tap water to saturation, autoclaved for 1 h, cooled to room temperature, and used to fill pots. The pots were 72-well TLC Square Plug trays (International Greenhouse Company, Danville, IL, USA), placed inside a tray with holes, then both placed inside another tray without holes (Hummert International). To prepare for sowing, the tray was filled with 2.5 L of fertilizer solution (0.01% Peters 20:20:20 (Hummert International) in tap water), and the 72-well tray was soaked until sowing was completed.

When sowing, a toothpick was used to place four seeds, evenly spaced, near the center of each well. After sowing, each tray was drained, covered with a propagation dome (Hummert International), and kept at 4 °C for 2 days before transfer to growth conditions (21 °C, 60% humidity, 14 h light/10 h dark, light at 80–100 µE m^−2^s^−1^). On day 9, counting from the time the tray was transferred to 21 °C, the propagation dome was removed. On day 11, plants were thinned so that only one plant remained. Trays were watered by sub-irrigation once a week. On day 20, trays were irrigated with the 0.01% fertilizer solution.

### 3.3. Plant Phenotype Analysis

Ion leakage was measured on harvested leaves 5 and 6 over a 2 h period as described by Vu et al. [25] and reported as percent of total ions. Plant leaf numbers were counted on live plants and checked by examination of photographs. To obtain the dry mass of rosettes, the rosettes were harvested, dried overnight in an oven at 105 °C, and weighed.

### 3.4. Experimental Design of the Main Heating Experiment

For each of three experimental blocks (replicate experiments), six plants of wild-type Columbia-0 (Col-0) accession and three plants of mutant *opr3* were grown in randomized positions in each of 17 72-well plug trays along with 63 other plants that were not further analyzed. Each plant was analyzed separately.

At 28 days of age each tray was treated with one of the temperature regimens (paths) shown in Figure 1 and harvested at one of the time points. All 17 treatments (17 trays) were repeated three times, with each replicate as an experimental block. Tray numbers and the corresponding treatments are listed in Appendix A. Randomization of plant position on the trays in each block was unique. Randomization was controlled so that for every three plants of identical genotype, at least 1 plant, but not more than 2 plants, was in an outside well. Each plant has a combined label including the block number, tray number, and well number (e.g., Appendix A). Trays were planted and cared for on a staggered schedule so that each step could be completed on the correct day of growth. 

### 3.5. Heating Treatments

Plants in the main heating experiment, involving wild-type and *opr3* mutants, were heat-treated using the regimen shown in Figure 1. Plants were subjected to temperature treatments beginning on day 28. For the severe heating experiment, the plants were subjected to an overnight treatment at 45 °C. The treatment began 1 h before the start of the dark period and ended 1 h after the dark period. In the severe heating experiment, the plants used for lipid analysis and those used to determine the number of leaves and rosette mass after heating were different trays of plants. In all cases, changes in temperatures were brought about by moving the plants among growth chambers pre-equilibrated to the appropriate temperature. 

### 3.6. Plant Sampling and Lipid Extraction

In the main heating experiment, after quickly harvesting leaves 5 and 6 from each plant for ion leakage measurements, the rest of the rosette was transferred to a 20 mL vial containing 4 mL of isopropanol with 0.01% butylated hydroxytoluene (BHT), preheated to 75 °C. Leaf number was determined as described by Telfer et al. [51]. After 15 min at 75 °C, the vials containing the plant materials in isopropanol were allowed to cool to room temperature before being stored at −80 °C. The lipid extraction was similar to that described by Vu et al. [13]. To begin lipid extraction, samples were allowed to warm to room temperature. To each vial, 12 mL of extraction solvent (chloroform: methanol: 300 mM ammonium acetate in water, 30: 41.5: 3.5, *v/v/v*) was added. The vials were shaken on an orbital shaker at 100 rpm for 24 h. For plants subject to the severe heating treatment, leaf 4, rather than the rosette, was sampled for lipid analysis. The extraction method described by Shiva et al. [52] was employed. Briefly, each leaf was harvested into 1.5 mL isopropanol with 0.01% BHT, preheated to 75 °C, and heated for 15 min. After cooling, 4.5 mL chloroform: isopropanol: methanol: water (30/25/41.5/3.5, *v/v/v/v*) was added, resulting in 6 mL of extract for each sample. The samples were shaken for 24 h.

After shaking, the extracted rosette or leaf from each vial was removed and put in an empty vial with the same label. The original vials with solvent were stored at −20 °C. Each extracted rosette or leaf in a non-capped vial was dried first in a fume hood for 1–2 h and then in an oven at 105 °C overnight. The dried rosettes were allowed to cool to room temperature. To eliminate electrostatic forces resulting from drying of the plant material, the materials were passed through an anti-static U ionizer (Haug, Germany). The dried plant material was weighed on a Mettler-Toledo AX balance (Mettler-Toledo, Greifensee, Switzerland). 

### 3.7. Sample Preparation for Mass Spectrometry. 

A mixture of 22 internal standards in chloroform, described by Vu et al. [13], was included in all mass spectrometry samples (including the sample vials, internal standard-only vials, and the quality control (QC) vials). A QC pool was prepared by pooling aliquots of all samples, adding internal standard mixture, adjusting the concentration to 0.0286 mg dry weight/mL, and aliquoting to mass spectrometry vials. QC sample vials were stored at −80 °C and brought to room temperature 1 h before analysis. The QC samples, all identical, were analyzed as every third or fourth sample throughout the analysis. QC samples were used to normalize the analysis across the samples, and to calculate the coefficient of variation for each analyte, as described previously [13,53]. Appendix A lists the positions of mass spectrometry vials in mass spectrometer autosampler racks in the main experiment.

To prepare the experimental samples, vials containing extracted total lipids were brought to room temperature. For the severe heating experiment, 100 µL of 0.63 M ammonium acetate was added to each 6 mL sample extract and vortexed for 1 min. No additions were made at this point to the samples from the main experiment. For both experiments, 20 µL of internal standard mixture was added to each 2 mL amber mass spectrometry sample vial. A volume equivalent to 0.04 mg leaf or rosette dry mass was added from each extract to its corresponding 2 mL amber vial. For samples in the main experiment, mass spectrometry solvent (isopropanol: chloroform: methanol: 0.3 M ammonium acetate in water, 25:30:41.5:3.5, *v/v/v/v*) was added to make the total volume 1.4 mL. For samples in the severe heating experiment, a mixture of isopropanol with 0.01% BHT/chloroform/methanol/0.2 M ammonium acetate in water (25/30/41.5/5.2, *v/v/v/v*) was added to bring the volume to 1.4 mL.

### 3.8. Lipid Analysis by Mass Spectrometry

Data were acquired on an ESI-triple quadrupole MS (Waters Xevo TQS, Waters Corporation, Milford, MA, USA) as described by Vu et al. [13]. Collision energies are listed in Appendix A. Other parameters specific to the analytes are provided in Appendix A of Vu et al. [13].

### 3.9. Data Processing, Unsaturation Index, Chain Length Index, and Statistical Analysis

Data from experimental samples were processed and normalized to QC samples as described by Vu et al. [13]. Internal standards used for each analyte are listed in Appendix A. 

To calculate the unsaturation index of each class, the molar amount of each lipid molecular species was multiplied times its number of double bonds (excluding double bonds in the fatty acid carbonyl groups). These values for all lipid molecular species in a class were summed, then divided by the total molar amount of the class and the number of acyl chains per molecule characteristic of the class to give the unsaturation index. Length index was calculated in the same way, substituting the combined acyl chain length of each molecular species for the number of double bonds.

For comparison of levels of lipid classes or the unsaturation or length index of classes at a single time point in the experiment (Figure 2, Figure 8, and Figure 9 and Appendix A; Appendix A), one-way ANOVA was conducted with Tukey’s multiple comparisons test, adjusted for FDR [54] using Metaboanalyst [55,56]. Mutant and wild-type phenotype data in Table 1 and Table 2 and Appendix A were compared by Student’s t-test.

For analysis of individual lipid species through the time course in the main experiment (Figure 3, Figure 4, Figure 5, Figure 6 and Figure 7 and Appendix A), a mixed effects ANOVA model was fitted to the data, and pairwise comparisons were made between paths. In the model, each of the three replicate parts of the experiment served as a block that contained 17 temperature treatments. Each of the three paths was formed through a specific combination of some of the 17 temperature treatments (Figure 1). Hence, in the comparison of Paths 1, 2, and 3, a fixed effect of path was included plus a fixed effect of the block to account for any systematic bias due to blocking. A random intercept was included for each temperature treatment nested in blocks. The *p*-values for all effects from all three pairwise comparisons were recorded, and adjusted *p*-values based on FDR control were calculated. 

All graphs were prepared using Origin, Version 2019b (Origin Lab Corporation, Northampton, MA, USA). To create the heatmap in Figure 2b, the data were autoscaled using Metaboanalyst [55,56], before creating the graph in Origin. Autoscaling is performed by dividing the difference between each lipid’s intensity and the mean intensity for that lipid across all samples by the standard deviation of the intensity across all samples. 

## 4. Conclusions

This study confirmed the leaf lipid changes described by others for heat stress in Arabidopsis, including increases in acMGDG, polygalactosylated galactolipids, PA, TAG, SG, and ASG. We also identified increases in several lipids not previously reported to increase under heat stress, including oxidized SQDG, oxidized PG, oxidized galactolipids, oxidized acMGDG, MGMG, and DGMG. Our work supports the notion that, during heat stress, unsaturated fatty acids are removed from galactolipids and incorporated into extra-plastidic phospholipids and TAG. We tested the hypothesis that deficiency in SG and ASG leads to defects in the formation of viable leaves after severe heat stress, but the results from our large-scale experiment failed to support this postulate. Figure 10 summarizes the details of the observed lipid changes under moderate and severe heat stress, along with current knowledge of the reactions involved in the heat response pathway. 

## Figures and Tables

**Figure 1 plants-09-00845-f001:**
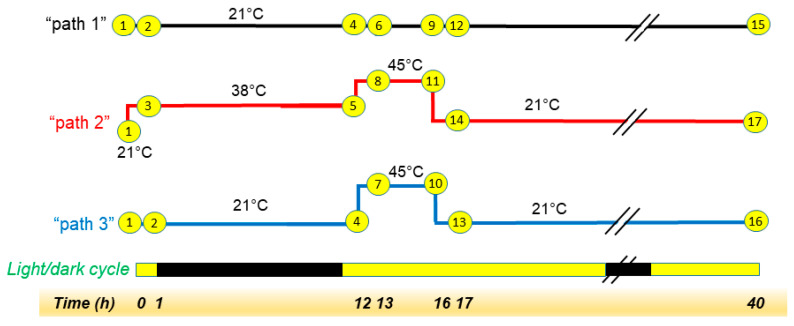
Heating experiment design. Plants 28-days-old were grown on a 14-h light, 10-h dark cycle at 21 °C. The experiment began 13 h into the light phase of the cycle (1 h before the start of the dark phase), and the original light cycle continued through the experiment. Plants were subjected to one of three treatments: “Path 1” (control): 21 °C for 40 h; “Path 2”: 38 °C for 12 h, 45 °C for 4 h, and 21 °C for 24 h; or “Path 3”: 21 °C for 12 h, 45 °C for 4 h, and 21 °C for 24 h. Sampling points were at 0 h, 1 h, 12 h, 13 h, 16 h (end of heat phase for paths 2 and 3), 17 h, and 40 h. At these times, leaves 5 and 6 were removed, and the remainder of the rosette was harvested and extracted for lipid analysis. (Each plant produced one experimental sample.) The plant tray numbers (Appendix A) corresponding to each treatment/time point are indicated in the yellow symbols.

**Figure 2 plants-09-00845-f002:**
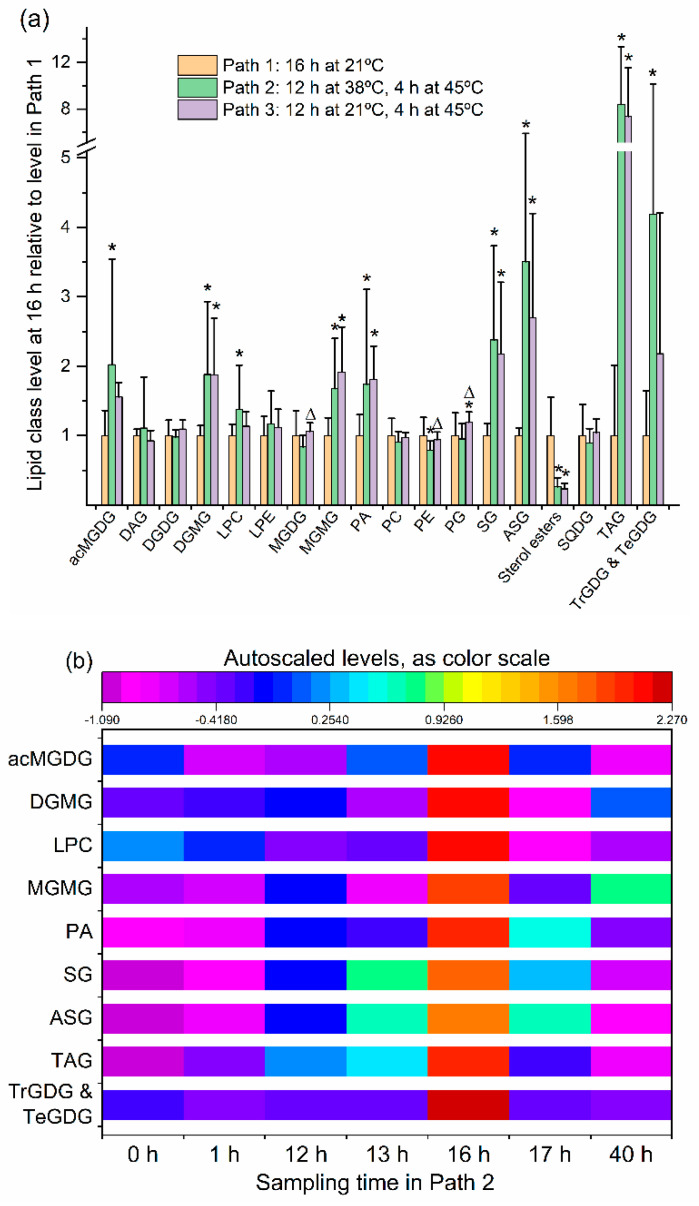
Levels of lipid head group classes in Arabidopsis leaves at the end of the heating treatment, compared to plants at 21 °C, and Path 2 lipid class time course. (**a**) Levels of lipid head group classes at the end of the heating treatment (16 h time point) in Path 2 (12 h at 38 °C and 4 h at 45 °C) and Path 3 (12 h at 21 °C and 4 h at 45 °C), relative to level at same time point in Path 1 (16 h at 21 °C). Asterisks indicate a significant difference in lipid level in Path 2 or 3 (by comparison with Path 1). Triangles indicate a difference of Path 3 from Path 2. Differences were evaluated by one-way ANOVA with Tukey’s multiple comparisons test, adjusted for FDR. A *p* value < 0.05, after correction for FDR, was considered significant. Error bars indicate standard deviation. (**b**) Heat map for lipid samples from the time course listing the nine lipid classes significantly increased in Path 2, based on autoscaled values for each lipid (see Section 3).

**Figure 3 plants-09-00845-f003:**
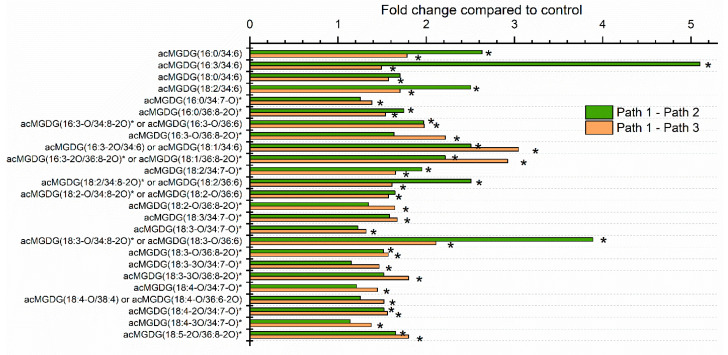
Leaf acylated monogalactosyldiacylglycerols, significantly altered by heat treatment. Values represent the level of each lipid under heat treatment over the level in control plants at 16 h after the start of the experiment. Control plants were maintained at 21 °C (Path 1), while Path 2 plants were subjected to 12 h at 38 °C plus 4 h at 45 °C, and Path 3 plants were subjected to 12 h at 21 °C plus 4 h at 45 °C. Lipid species with names ending in an asterisk are likely to contain oxophytodienoic acid (OPDA). A mixed effects ANOVA model was fitted to the data, and pairwise comparisons were made between paths; *p*-values of <0.05, after adjustment for FDR, were considered significant. The lipid species shown were those significantly altered in Path 2 and/or 3 compared to Path 1, as indicated by asterisks within the bar graph.

**Figure 4 plants-09-00845-f004:**
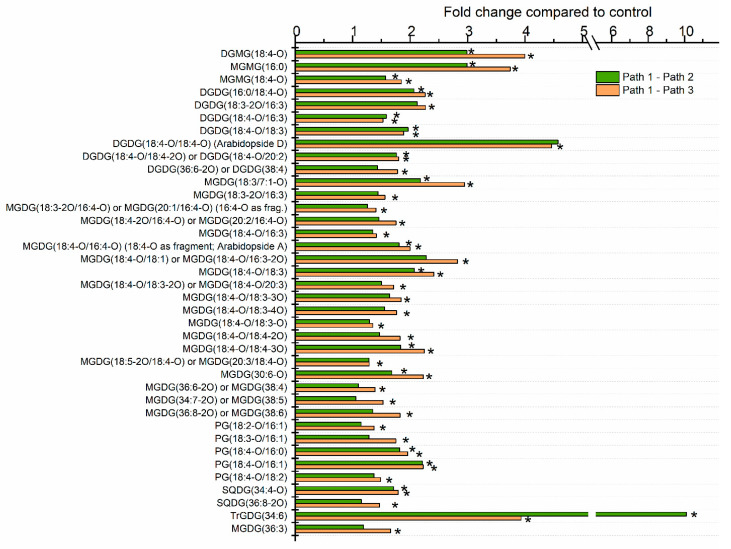
Plastidically-formed oxidized di- and monoacylglycerol leaf lipids, significantly altered by heat treatment. Values represent the level of each lipid under heat treatment over the level in control plants at 16 h after the start of the experiment. Control plants were maintained at 21 °C (Path 1), while Path 2 plants were subjected to 12 h at 38 °C plus 4 h at 45 °C, and Path 3 plants were subjected to 12 h at 21 °C plus 4 h at 45 °C. A mixed effects ANOVA model was fitted to the data, and pairwise comparisons were made between paths; *p*-values were adjusted for FDR. The lipid species shown were those significantly altered in Path 2 and/or 3 compared to Path 1, as indicated by asterisks on the bar graph.

**Figure 5 plants-09-00845-f005:**
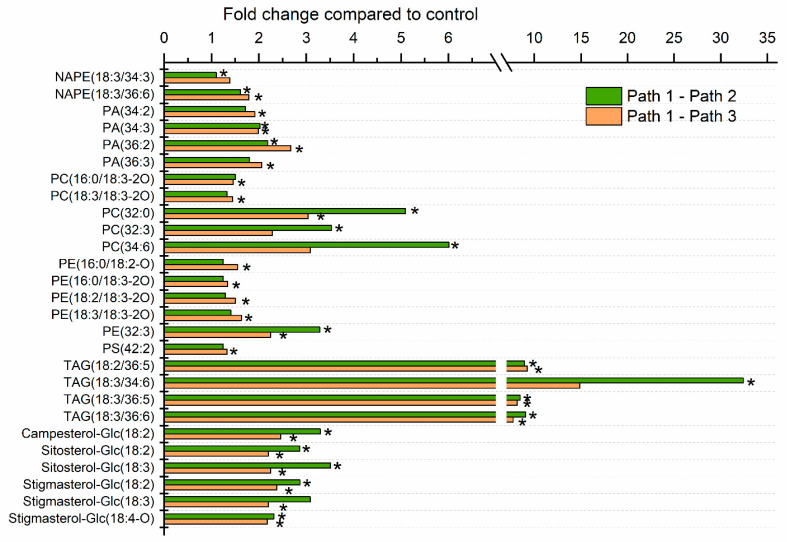
Extra-plastidic leaf phospholipids, acyl sterol glucosides, and triacylglycerols, significantly altered by heat treatment. Values represent the level of each lipid under heat treatment over the level in control plants at 16 h after the start of the experiment. The control plants were continually subjected to 21 °C (Path 1), while Path 2 plants were subjected to 12 h at 38 °C plus 4 h at 45 °C, and Path 3 plants were subjected to 12 h at 21 °C plus 4 h at 45 °C. A mixed effects ANOVA model was fitted to the data, and pairwise comparisons were made between paths; *p*-values were adjusted for FDR. The lipid species shown were those significantly altered in Path 2 and/or 3 compared to Path 1, as indicated by asterisks.

**Figure 6 plants-09-00845-f006:**
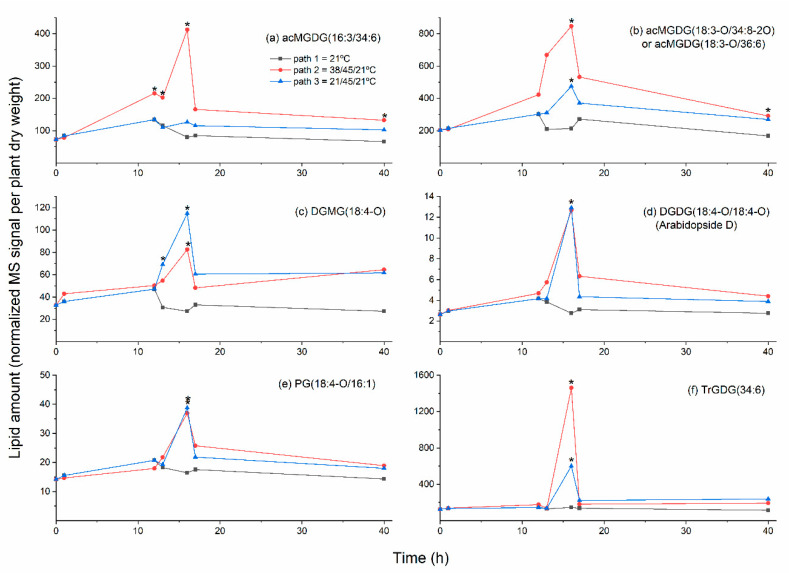
Levels of leaf plastid-synthesized lipids as function of time during control and heat stress treatments. Treatment details are shown in Figure 1. (**a**) acMGDG(16:3/34:6); (**b**) acMGDG(18:3-O/34:8-2O) or acMGDG(18:3-O/36:6); (**c**) DGMG(18:4-O); (**d**) DGDG(18:4-O/18:4-O), i.e., Arabidopside D; (**e**) PG(18:4-O/16:1); (**f**) TrGDG(34:6). A mixed effects ANOVA model was fitted to the data, and pairwise comparisons were made between paths; *p*-values were adjusted for FDR. An asterisk indicates that the value is significantly different from the control (Path 1) value for that time point (*p* < 0.05), using a *p*-value adjusted for false discovery rate (FDR; Appendix A).

**Figure 7 plants-09-00845-f007:**
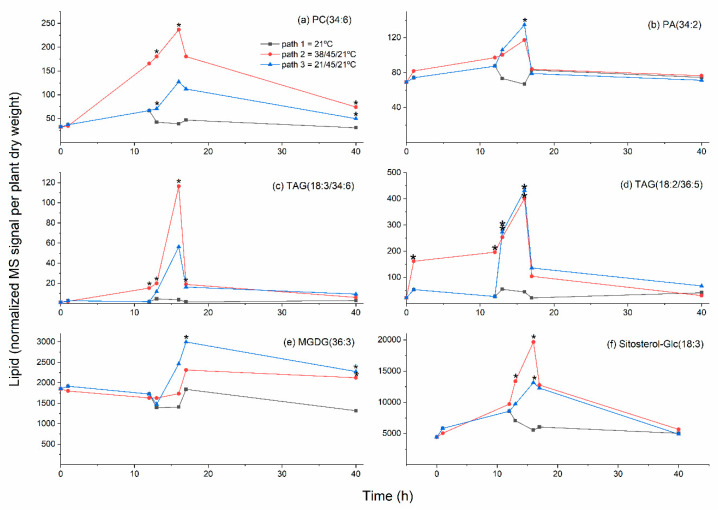
Levels of additional lipid species as a function of time during control and heat stress treatments. Treatment details are shown in Figure 1. (**a**) PC(34:6); (**b**) PA(34:2); (**c**) TAG(18:3/34:6); (**d**) TAG(18:2/36:5); (**e**) MGDG(36:3); (**f**) Sitosterol-Glc(18:3). A mixed effects ANOVA model was fitted to the data, and pairwise comparisons were made between paths; *p*-values were adjusted for FDR. An asterisk indicates that the value is significantly different from the control (Path 1) value for that time point (*p* < 0.05), using a *p*-value adjusted for false discovery rate (FDR; Appendix A).

**Figure 8 plants-09-00845-f008:**
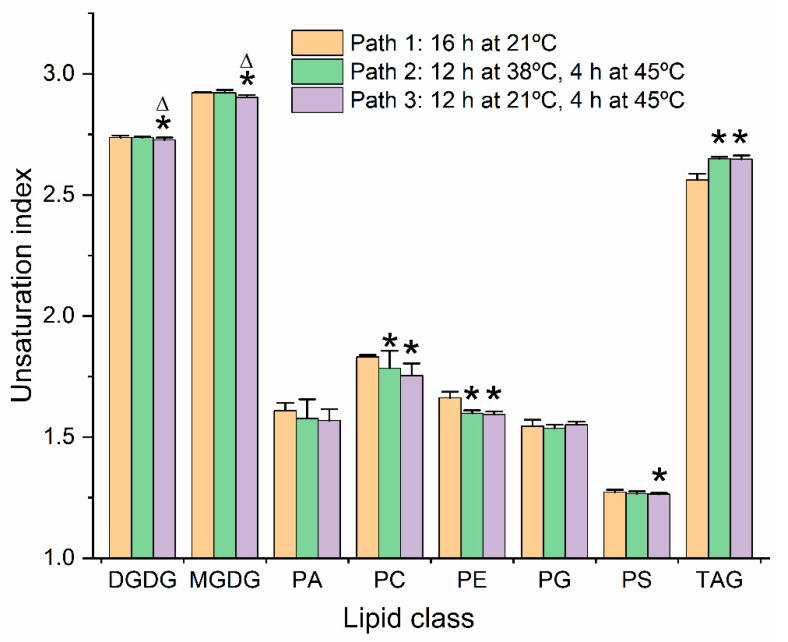
Unsaturation indices of control and heat-treated diacyl lipid species from leaves of wild-type *Arabidopsis thaliana*. Unsaturation indices indicate the average number of double bonds per acyl chain in each lipid class, calculated as described in Materials and Methods. Comparisons were made at the 16 h time point for control plants (Path 1, orange bars), plants treated for 12 h at 38 °C followed by 4 h at 45 °C (Path 2, green), and 12 h at 21 °C followed by 4 h at 45 °C (Path 3, lavender). Asterisks indicate significant differences from that of the Path 1 control plants (*p* < 0.05) by one-way ANOVA, with Tukey’s multiple comparisons test, adjusted for FDR. Triangles indicate that a Path 3 value is significantly different than a Path 2 value. Error bars indicate standard deviation.

**Figure 9 plants-09-00845-f009:**
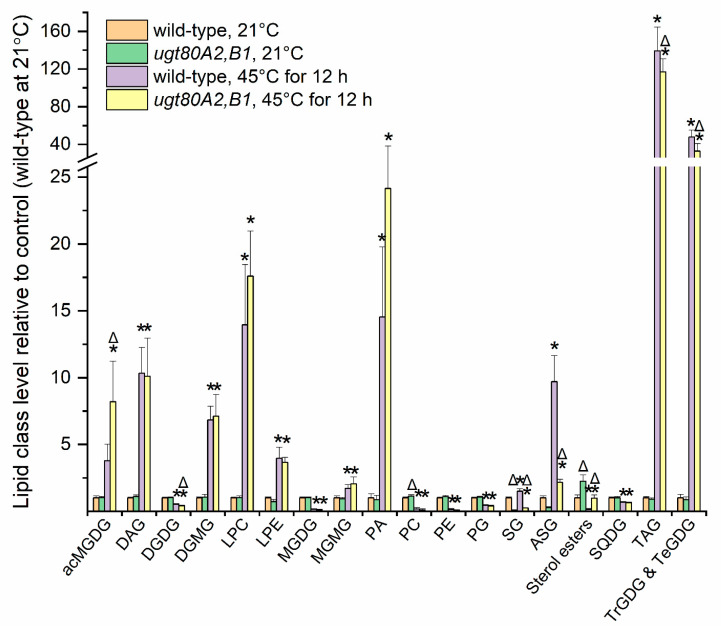
Levels of lipid head group classes in wild-type and *ugt80A2,B1* double mutant Arabidopsis leaves after severe heating treatment of 12 h at 45 °C. Asterisks indicate a significant difference in lipid level due to the heating treatment (compared with the same genotype held at 21 °C). Triangles indicate a difference in *ugt80A2,B1* double mutant plants versus wild-type plants under the same treatment conditions. Differences were evaluated by one-way ANOVA with Tukey’s multiple comparisons test, adjusted for FDR. A *p*-value < 0.05, after correction for false discovery rate, was considered significant. Error bars indicate standard deviation.

**Figure 10 plants-09-00845-f010:**
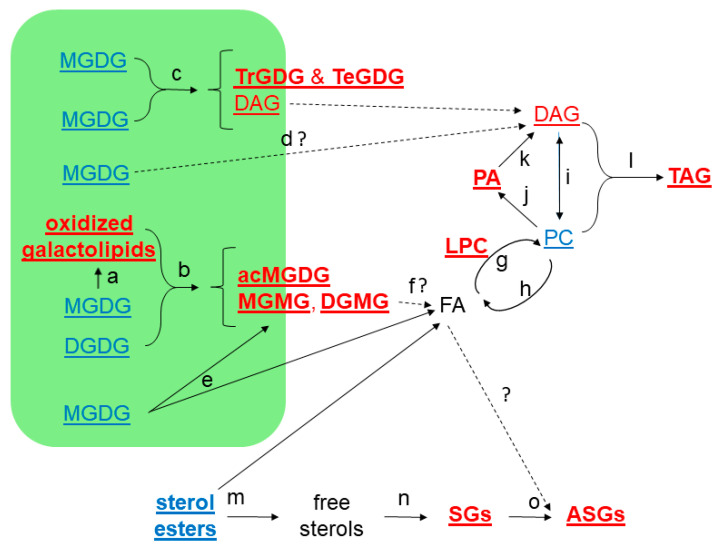
Lipid metabolic pathways involved in heat stress response in Arabidopsis leaves. Lipid groups in bold type were altered significantly under the moderate heat stress conditions of Path 2 of the main experiment (Figure 2). Lipid groups underlined were changed under severe stress (Figure 9). The green area represents the chloroplast and its lipids. Lipid groups in red type are increased, those in blue are decreased, and those in black were not measured. The letters “a” to “o” indicate reactions. Dashed lines indicate processes that are less well understood, or less clearly involved in producing the observed lipid changes, in comparison to the solid lines. “a”: Reactions involved in oxidation of galactolipids. The most well-characterized process involves formation of OPDA (or dnOPDA) through a lipoxygenase, allene oxide synthase, allene oxide cyclase, and oxophytodienic acid reductase. This conversion of 18:3 to OPDA (or 16:3 to dnOPDA) was demonstrated to occur while the fatty acid is esterified to the galactolipid [26]. The processes forming other oxidized galactolipids during heat stress in Arabidopsis leaves are less well-characterized, but they are likely to include the non-enzymatic formation of phytoprostanes [57], as well as other enzymatic pathways. “b”: Acylation of MGDG on the 6-position of the galactose ring with a fatty acid coming from DGDG or a second MGDG, catalyzed by AGAP1 [24]. “c”: Processive galactosylation of MGDG, catalyzed by the galactolipid galactosyltransferase SFR2, to form polygalactolipids, such as TrGDG and TeGDG, and DAG [29,30]. “d”: Possible formation of DAG from galactolipids by an unknown lipase. This pathway could contribute 16:3 as 18:3/16:3 DAG to the PA, PC, and TAG pools. “e”: Fatty acids can be hydrolyzed from MGDG by acylhydrolases, such as HEAT-SENSITIVE LIPASE, which cleaves the fatty acid (typically 18:3) in the 1-position of MGDG [40]. “f”: MGMG or DGMG could be hydrolyzed to release fatty acids, but the identity of the gene product catalyzing this reaction is not known. “g” and “h”: The acyl editing pathway for incorporation into and removal of fatty acids from PC. “g” represents LPCAT, which transfers a fatty acid (after activation to acylCoA) to LPC to form PC [41], whereas “h” represents an acylhydrolase, acting on PC, resulting in LPC and a fatty acid, or perhaps a reverse LPCAT reaction, resulting in LPC and fatty acyl CoA [42,58]. The acyl editing cycle can bring new fatty acids into PC. Note that an LPC:LPC transacylase (LPC + LPC to PC + glycerophosphocholine) that could contribute to PC formation via acyl editing has been identified in safflower seeds [42,59]. “i”: DAG and PC can be interconverted, although when heating stress is severe, PC levels drop while DAG levels rise. PC can be hydrolyzed by a phospholipase C to form DAG, whereas the enzyme that synthesizes PC from DAG, CTP-choline:DAG phosphocholine transferase, can also catalyze the reverse reaction [42,60]. “j” and “k”: DAG can also be formed from PC via a phospholipase D, possibly by PLDδ [36], followed by a PA phosphatase. “l”: TAG is formed during heating by transfer of a fatty acyl chain from PC to DAG, as catalyzed by PDAT1 [7]. “m”: Hydrolysis of sterol esters to free sterols and fatty acids. Although sterol ester levels drop during heating, the levels of sterol esters in Arabidopsis are low, so this is a minor source of fatty acids and free sterols. “n”: Glucosylation of sterols by transfer of glucose from UDP-glucose by UGT80B1 and UGT80A2, which account for 85–90% of the SG formed [46,48]. “o”: Acylation of SGs, from an unknown acyl donor, to form ASGs, catalyzed by an unknown enzyme.

**Table 1 plants-09-00845-t001:** Leaf viability in wild-type versus *ugt80A2,B1* double mutant plants in response to a 12-h heat treatment at 45 °C at 30 day (d) of age. The number of rosette leaves of untreated plants was determined at 30 d and 42 d. Leaves of treated plants were counted 12 d after the onset of the 12 h heat treatment. Student’s *t*-test indicated that the untreated double mutant plants had fewer leaves at 42 d than wild-type plants (*p* < 0.05). However, the number of leaves in heat-treated wild-type and *ugt80A2,B1* double mutant plants was not significantly different.

	*n* (Number of Plants of Each Line)	Number of Viable Rosette Leaves	
		Wild-Type	*ugt80A2,B1* Double Mutant
Untreated, grown at 21 °C (30 d)	324	9.9 ± 1.7	9.8 ± 1.6
Untreated control, grown at 21 °C (42 d)	103	18.3 ± 1.4	17.8 ± 1.4 *
Treated at 45 °C for 12 h at 30 days, otherwise grown at 21 °C (42 d)	108	10.9 ± 3.3	10.5 ± 3.2

**Table 2 plants-09-00845-t002:** Dry mass of rosettes of wild-type and *ugt80A2,B1* plants in response to severe heat stress at 30 d of age. Dry masses of untreated rosettes were determined after harvest at 30 d and 42 d. Treated rosettes were harvested 12 d after the onset of a 12 h, 45 °C heat treatment. The dry masses of wild-type and *ugt80A2,B1* double mutant rosettes were not significantly different under any condition, as evaluated by Student’s *t*-test (*p* < 0.05).

	*n* (Number of Plants of Each Line)	Dry Mass of Rosette (g)	
		Wild-Type	*ugt80A2,B1* Double Mutant
Untreated, grown at 21 °C (30 d)	103	0.08 ± 0.03	0.08 ± 0.05
Untreated control, grown at 21 °C (42 d)	103	0.27 ± 0.05	0.26 ± 0.05
Treated at 45 °C for 12 h at 30 days, otherwise grown at 21°C (42 d)	108	0.10 ± 0.04	0.10 ± 0.04

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
