# Peer review of "Leaf Lipid Alterations in Response to Heat Stress of Arabidopsis thaliana"

_plants, 2020, doi:10.3390/plants9070845_

Round 1
Reviewer 1 Report
The manuscript “Leaf lipid alterations in response to heat stress of Arabidopsis thaliana” provides a greatly enhanced analysis of changes in lipid molecular species due to heat stress. This includes lipids not previously analyzed in response heat stress. This analysis will help us to understand the molecular mechanisms that plants use to respond to changing temperatures.
Overall the experiments appear to be of high quality, and provide useful information.
A few specific comments are below:
Lines 370-373. Are you suggesting an sn-2 lipase for MGDG, or complete turnover of MGDG that could release both acyl chains? If complete acyl group turnover is present, is there any indication of increased synthesis of MGDG that might replace the MGDG lost?
Lines 373-374. In regards to the speculation of an acyl transferase that adds acyl groups to PC in the outer chloroplast membrane. Recent work in Arabidopsis lpcat1 lpcat2 mutant has demonstrated that the LPCATs are localized to the chloroplast and are involved in putting fatty acids exported from the chloroplast into PC. See: Karki N, Johnson BS, Bates PD (2019) Metabolically Distinct Pools of Phosphatidylcholine Are Involved in Trafficking of Fatty Acids out of and into the Chloroplast for Membrane Production. Plant Cell 31: 2768-2788
Section 2.7. Unsaturation index has been used many times, especially when only GC of FAMEs is the analysis method. However, with MS you have much more detail that might provide more informative analysis than just total number of double bonds in each lipid. What about number of double bonds per molecular species, or changes in the number of carbons per molecular species? It would be great if a more detailed analysis of the molecular species specific type changes gained from MS analysis could provide more details to what lipid molecular species might lead to thermotolerance than just unsaturation index.
Reviewer 2 Report
Shiva and colleagues characterize leaf lipids in response to heat stress in A. thaliana in the submitted manuscript. The sampling schemes used were logical and allowed clear comparisons between timepoints/treatments for specific analytes. The manuscript is well-written, and all concerns are minor.
The Result and Discussion subheadings 2.1 and 2.2 describe the experimental method and analyses rather than a result.
Figure 1 is an extremely useful guide to the experimental design.
It is hard to understand whether leaves were pooled for mass spec from the manuscript text. How does the 18 plants (line 113) from each timepoint fit in? Please clarify this in the text or methods.
What are the error bars for Fig 2a, 9, and S2 representing? There are also some graphical artifacts in Fig 2A where vertical or horizontal parts of the error bars are missing. Also consider adding Path 1, 2, 3 into the 2A legend and y-axis label.
In Fig 2B, is there a way to have separation between each row to discourage the reader from comparing across lipid classes since those comparisons are invalid (based on lines 124-126)?
In line 157, can you confirm in the text whether the 13 analytes at the 1-hour timepoint were only different in path 2?
In regard to the analysis, it may be easier for readers to understand that the split-plot ANOVA was used for the analysis of the repeated measures; please clarify this in the methods that main factor was the different Paths and the “subplots” were the different times.
Add a reference to figure 3 in line 216.
